# Detection of new pioneer transcription factors as cell-type-specific nucleosome binders

**Yunhui Peng[1,2]\*[†], Wei Song[2], Vladimir B Teif[3], Ivan Ovcharenko[2], David Landsman[2], Anna R Panchenko[4,5,6,7]\***

[1]Institute of Biophysics and Department of Physics, Central China Normal University, Wuhan, China; [2]National Library of Medicine, National Institutes of Health, Bethesda, United States; [3]School of Life Sciences, University of Essex, Wivenhoe Park, Colchester, United Kingdom; [4]Department of Pathology and Molecular Medicine, Queen's University, Kingston, Canada; [5]Department of Biology and Molecular Sciences, Queen's University, Kingston, Canada; [6]School of Computing, Queen's University, Kingston, Canada; [7]Ontario Institute of Cancer Research, Toronto, Canada

**\*For correspondence:**
yunhuipeng@ccnu.edu.cn (YP);
anna.panchenko@queensu.ca
(ARP)

**Present address:** [†]Institute of Biophysics and Department of Physics, Central China Normal University, Wuhan, China

**Competing interest:** The authors declare that no competing interests exist.

**Abstract** Wrapping of DNA into nucleosomes restricts accessibility to DNA and may affect the recognition of binding motifs by transcription factors. A certain class of transcription factors, the pioneer transcription factors, can specifically recognize their DNA binding sites on nucleosomes, initiate local chromatin opening, and facilitate the binding of co-factors in a cell-type-specific manner. For the majority of human pioneer transcription factors, the locations of their binding sites, mechanisms of binding, and regulation remain unknown. We have developed a computational method to predict the cell-type-specific ability of transcription factors to bind nucleosomes by integrating ChIP-seq, MNase-seq, and DNase-seq data with details of nucleosome structure. We have demonstrated the ability of our approach in discriminating pioneer from canonical transcription factors and predicted new potential pioneer transcription factors in H1, K562, HepG2, and HeLa-S3 cell lines. Last, we systematically analyzed the interaction modes between various pioneer transcription factors and detected several clusters of distinctive binding sites on nucleosomal DNA.

## eLife assessment

This **valuable** study aims to identify pioneer transcription factors - which are defined as transcription factors that compete with nucleosomes for DNA binding. The authors provide methods for identifying pioneer transcription factors on a cell type basis, using nucleosome positioning and motif information across different cell lines. The evidence to support the claims is largely **solid**. This work will be of interest to computational and molecular biologists working on transcription factors.

## Introduction

In eukaryotic cell, DNA is packaged in the form of chromatin, yet, it should be dynamically accessed during transcription and replication processes at high spatiotemporal precision (*Isbel et al., 2022*). Open chromatin is thought to comprise actively transcribed genes, while compact chromatin contains repressed genes. However, many recent observations point to a limited association between DNA accessibility, chromatin compaction, and gene transcription at the global genomic scale. Indeed, rapid transcription activation may occur by relatively small changes of DNA solvent exposure, and localized nucleosomal array structures can be dynamically exposed without the large-scale chromatin

rearrangements (*Zaret, 2020*; *Iwafuchi-Doi, 2019*; *Kono and Ishida, 2020*; *Chereji et al., 2019*). Nucleosomes represent the basic subunits of chromatin structure and function. They comprise a histone octamer of four types of core histones of two copies each, and ~147 bp of DNA wrapped around them (*Luger et al., 1997*). Wrapping of DNA into nucleosomes inherently restricts DNA accessibility and the recognition of binding motifs by transcription factors (TFs). Intrinsically disordered histone tails flank histone core domains and may also modulate DNA accessibility by forming transient interactions with the nucleosomal and linker DNA (*Peng et al., 2021a*). The control of the DNA accessibility at nucleosomal and subnucleosomal scales is of major importance in understanding of how certain TFs can target compact chromatin to induce transcription activation or repression (*Klemm et al., 2019*; *Kornberg and Lorch, 2020*; *Peng et al., 2021b*).

The differentiation of cells into different lineages occurs through chromatin reprogramming, involving the cooperative behavior of various TFs (*Peñalosa-Ruiz et al., 2019*; *Balsalobre and Drouin, 2022*). Although nucleosomes generally hinder the binding of TFs, a certain class, so-called pioneer transcription factors (PTFs), can specifically recognize their binding sites on nucleosomal DNA, in some cases initiating local chromatin opening and facilitating subsequent binding of other co-factors in a cell-type-specific manner (*Zaret, 2020*; *Balsalobre and Drouin, 2022*). Several studies have revealed the critical roles of PTFs in mediating the cell-type-specific gene expression and establishment of cell lineage reprogramming (*Peñalosa-Ruiz et al., 2019*; *Balsalobre and Drouin, 2022*; *Jaenisch and Young, 2008*).

Significant efforts have been made to characterize the interaction landscape between various TFs and nucleosomes (*Zhu et al., 2018*; *Meers et al., 2019*; *Soufi et al., 2015*; *Luzete-Monteiro and Zaret, 2022*; *Tan and Takada, 2020*). Using the NCAP-SELEX approach, a recent study has characterized the interaction modes between nucleosomes and 220 TFs (*Zhu et al., 2018*). Another high-throughput protein microarray study of 593 human TFs systematically identified the structural features of TFs binding with nucleosomes (*Fernandez Garcia et al., 2019*.) It has been revealed that the vast majority of TFs preferably bind naked DNA instead of nucleosomal DNA at physiological concentrations, whereas certain TFs specifically target nucleosomes at different locations and orientations (*Zhu et al., 2018*; *Meers et al., 2019*; *Soufi et al., 2015*). Moreover, several structures of PTFs in complex with nucleosomes have recently been resolved (*Min and Liu, 2021*; *Michael et al., 2020*; *Dodonova et al., 2020*; *Tanaka et al., 2020*). Despite significant advances in recent experimental approaches, the interaction modes of most TFs with nucleosomes remain obscure and their cell-type-specific pioneer activities are largely unknown. Development of new computational approaches can improve our understanding of binding properties of various TFs with nucleosomes, thereby helping to identify novel PTFs.

Through the advances in high-throughput sequencing techniques, a large volume of data has been generated (e.g. ChIP-seq, ATAC-seq, DNase-seq, and MNase-seq data; *Davis et al., 2018*), which allows us to gain insights into the chromatin structure and details of epigenetic binding events. The rapid growth of such datasets has further stimulated the development of computational methods to characterize cell-type-specific TFs (*Sherwood et al., 2014*; *Jankowski et al., 2016*; *Zheng et al., 2021*; *Avsec et al., 2021*). Several machine learning models have been proposed to predict TF binding sites and identify sequence context features critical for TF binding (*Sherwood et al., 2014*; *Zheng et al., 2021*; *Avsec et al., 2021*; *Kishan et al., 2021*). In addition, gene regulatory network-based approaches have helped to identify the key TFs in cell fate determination (*Xu et al., 2021*; *Heuts et al., 2022*). Despite the success of computational methods in genome-wide prediction of binding sites of canonical TFs, the locations of PTFs' binding sites, mechanisms of binding, and regulation have not been systematically explored.

Integrating data on nucleosome positioning and DNA accessibility (MNase-seq, ATAC-seq, and DNase-seq) with the data on DNA binding events, available through ChIP-seq and other methods, can reveal the interplay between TF binding and nucleosome positioning, providing insights into the mechanisms of PTF interactions with nucleosomes and their ability to modulate chromatin accessibility (*Teif et al., 2014*; *Chen et al., 2021*; *Yu et al., 2021*; *Gong et al., 2022*).Here we have developed a computational method to study the ability of TFs to bind nucleosomes by using ChIP-seq, MNase-seq, and DNase-seq datasets from five different cell lines. Our results point to the capability of our method to discriminate between pioneer and canonical TFs using experimental benchmarks. Additionally, we have predicted several TFs as potentially new cell-type-specific PTFs in H1, K562, HepG2, and

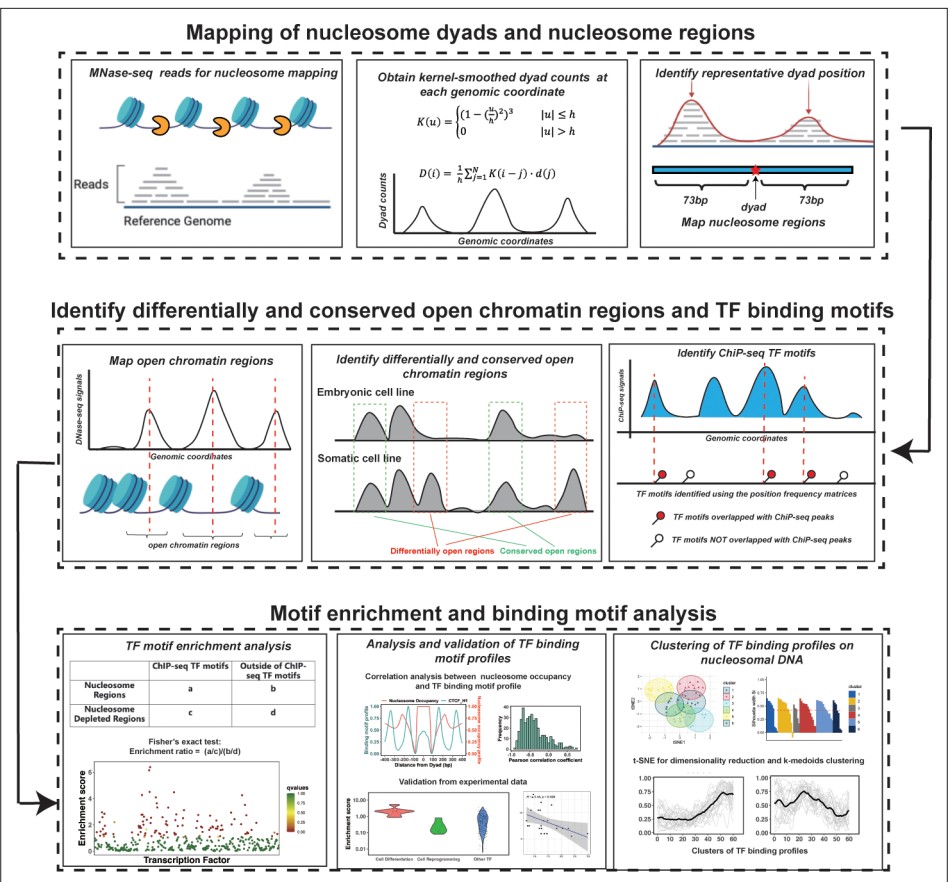

**Figure 1.** The computational framework to analyze the ability of transcription factors to bind to nucleosomes by integrating ChIP-seq, MNase-seq, and DNase-seq data for motif enrichment and binding motif analysis.

The online version of this article includes the following source data and figure supplement(s) for figure 1:

**Figure supplement 1.** Identification of differentially open and conserved open chromatin regions between H1 embryonic cell line and any other differentiated cell lines.

**Figure supplement 2.** Identification of dinucleotide patterns of nucleosomal DNA (MCF7 cell line is shown as a representative case).

**Figure supplement 2—source data 1.** Source data for dinucleotide patterns of nucleosomal DNA.

HeLa cell lines and performed multiple validations. Last, we have systemically analyzed the interaction modes between various PTFs and nucleosomes and detected six clusters of distinctive binding sites on nucleosomal DNA.

## Results

### TF binding motif enrichment analysis can identify PTFs

Nucleosomes are generally considered as an impediment to the binding of TFs to DNA and thus binding sites of TFs are typically depleted in DNA regions with high nucleosome occupancy. However, PTFs can recognize their binding motifs on nucleosomal DNA and trigger the opening of chromatin to recruit other TFs in a cell-type-specific fashion (*Zaret, 2020*; *Zaret and Mango, 2016*). Therefore, we hypothesized that DNA binding sites of PTFs should not be depleted and, in some cases, should be enriched on nucleosome regions (NRs). At the same time, those canonical TFs that can preferentially bind to the naked DNA should exhibit the depletion of binding sites on NRs and enrichment in nucleosome-free regions. To quantitatively assess these trends, we have performed binding motif enrichment analysis for each of the 225 TFs and calculated the motif enrichment scores (*Supplementary file 1—tables 1–4*). We have further analyzed the enrichment

of TF binding sites on NRs compared to the nucleosome-depleted regions (NDRs) (see Materials and methods for definition). The overall workflow of our computational framework is shown in *Figure 1*.

To validate our results, we have compiled three sets of known PTFs (*Zaret, 2020*; *Balsalobre and Drouin, 2022*; *Zhu et al., 2018*; *Soufi et al., 2015*) (which could also be found in our list of 225 TFs) as positives (*Supplementary file 1—tables 5 and 6*), whereas other TFs were considered as negatives for this test. Test set 1 includes 32 known PTFs. Some known PTFs were not included as they were not present in our original datasets. Test set 1 comprises Test set 2 and 3 and other known PTFs. Test set 2 includes 11 known PTFs with specific roles in cell differentiation. Test set 3 includes seven known PTFs critical for the maintenance of embryonic stem cells or the reprogramming of somatic cells into induced pluripotent stem cells (*Supplementary file 1—table 6*). As we show in the next section, the negative set may also contain PTFs, therefore the classification accuracy values provided below can be considered as a lower bound estimates. Then, we performed the enrichment analysis by calculating the binding motif enrichment of different TFs on nucleosomal regions compared to NDRs.

The enrichment score of the 32 known PTFs from Test set 1 was found to be significantly higher than for other factors (p-value = $2.12*10^{-7}$ for all TFs and p-value = $4.33*10^{-5}$ for highly expressed TFs, Mann-Whitney U test, *Figure 2—figure supplement 1* and *Supplementary file 1—table 7*). The results also show the efficiency of enrichment scores for the classification of PTFs (*Supplementary file 1—table 7*, receiver operating curve [ROC] AUC = 0.69, precision-recall [PR] AUC = 0.33 and maximal Matthews correlation coefficient [MCC] = 0.31 for all TFs and ROC AUC = 0.71, PR AUC = 0.37 and maximal MCC = 0.31 for significantly expressed TFs).

Next, the validation pertaining to the ability of PTFs to open the closed chromatin was performed. We hypothesized that the enrichment score calculated based on NRs located in differentially open chromatin regions and NDRs located in conserved open chromatin regions would perform best in the classification of PTFs essential for cell differentiation (Test set 2). As can be seen in *Figure 2* and *Supplementary file 1—table 7*, it is indeed the case and the classification accuracy increases from ROC AUC = 0.69 to 0.89 (from 0.71 to 0.92 for expressed TFs) upon the inclusion of differentially open regions compared to open regions (maximal MCC increased from 0.31 to 0.42). We found that known PTFs that acted as key regulators of cell differentiation had the highest enrichment scores in our ranking (*Figure 2c*). These cases (Test set 2) mainly included PTFs from the FOXA, GATA, and CEBP families (*Costa et al., 2003*; *Smale, 2010*; *Lee et al., 2005*) with GATA1 and GATA2 showing the highest enrichment scores.

Interestingly, PTFs in Test set 3 (responsible for the maintenance of embryonic stem cell or reprogramming of somatic cells) showed significantly lower enrichment scores compared to other TFs (*Figure 2c* and *Figure 2—figure supplement 2*). Yamanaka PTFs (POU5F1/OCT4 and KLF4) (*Takahashi and Yamanaka, 2006*) were strongly depleted at nucleosomes (*Figure 2c*). It has been previously shown that Yamanaka PTFs might recognize partial sequence motifs on nucleosomal DNA and require other factors for their binding to nucleosomes (*Soufi et al., 2015*) and therefore their enrichment score might not be expected to be high. We also found known PTFs with relatively low enrichment scores including NFYA, NFYB, NFYC, and ESRRB. These TFs regulate stem cell proliferation and maintenance of stem cell identity (*Bungartz et al., 2012*; *Gao et al., 2019*; *Iyer et al., 2016*; *Figure 2—figure supplement 2*).

However, when we repeated our analysis by redefining differentially open regions as those closed in differentiated cell lines and open in H1 embryonic cell line, then ESSRB and Yamanaka PTF POU5F1 (OCT4) showed significantly higher enrichment scores (*Figure 2—figure supplement 3*). This could be explained by the roles of Yamanaka factors in cellular reprogramming – they reprogram somatic differentiated cells into induced pluripotent stem cells.

Since PTFs often target enhancers, we have repeated the enrichment analysis using NRs located in differentially active enhancer regions and NDRs located in the conserved active enhancer regions but it led to the worse performance of the PTF classification (*Figure 2—figure supplement 4*). Here, differential enhancer regions refer to the active enhancer regions in an differentiated cell line that have less than 20% overlap with any active enhancer regions in embryonic (H1) cell line. Conserved active enhancer regions represent active enhancer regions that are more than 80% shared between embryonic H1 and at least one other differentiated cell line used in this study. We also explored different thresholds in defining *differentially open* and *conserved open chromatin regions* but the

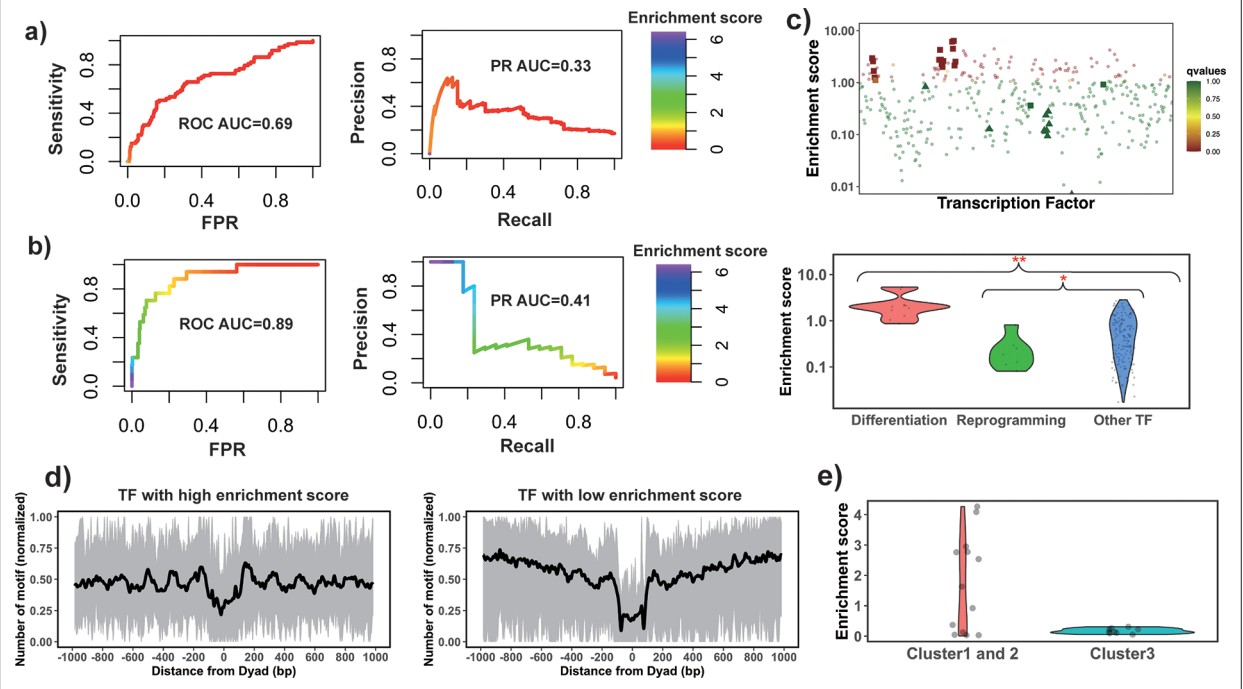

**Figure 2.** Identify pioneer transcription factors (PTFs) using motif enrichment analysis. (**a**) Transcription factor (TF) motif enrichment score is used to distinguish 32 known PTFs (Test set 1) from other TFs. Receiver operating curve (ROC) and precision-recall (PR) curve analysis of motif enrichment scores are performed. Here, nucleosome regions (NRs) were determined as genomic regions (147bp long) centered at the representative dyad positions and the nucleosome-depleted regions (NDRs) represent genomic regions free of nucleosomes and are located in open chromatin regions. The random retrieval classifier would predict with AUC = 0.5and PR = the fraction of true positives = 0.17. (**b**) TF motif enrichment score is used to distinguish 11 known PTFs with essential roles in cell differentiation (Test set 2) from other TFs. Here, NRs in differentially open and NDRs in conserved open chromatin regions are used in enrichment analysis. The random retrieval classifier would predict with AUC = 0.5and PR = 0.04. (**c**) Classification of PTFs by binding motif enrichment scores. Known PTFs from Test set 2 and Test set 3 are indicated by squares and triangles, while other TFs are shown as circles. Colors corresponds to false discovery rate (FDR) q-values. Mann-Whitney U tests are performed under the null hypothesis that PTF's mean values of enrichment scores are equal to canonical TFs. * – p-value <0.05; ** – p-value <0.005. (**d**) Binding motif profile of TFs with the highest and lowest motif enrichment scores (ranked at the top or bottom 10% among all TFs). The number of motifs for each TF is normalized within the range between 0 and 1 as follows: $X(i)_{normalized} = (X(i) - X_{min})/(X_{max} - X_{min})$, $X(i)$ is the number of sequences which have TF binding sites at the $i$th base pair from the nucleosomal dyad position; $X_{max}$ and $X_{min}$ represent the maximal and minimal counts of sequence fragments respectively. (**e**) Comparison of the enrichment score of TFs in different clusters identified from recent electromobility shift assay (EMSA) experiments (*Fernandez Garcia et al., 2019*). Only 13 TFs could be found in EMSA and our dataset. Clusters 1 and 2 include strong binders to both naked DNA and nucleosomal DNA and weak binders to both naked DNA and nucleosomal DNA (only one TF). Cluster 3: strong binders to naked DNA but weak binders to nucleosomal DNA.

The online version of this article includes the following source data and figure supplement(s) for figure 2:

**Source data 1.** Source data for all TF enrichment scores calculated from motif enrichment analysis.

**Figure supplement 1.** Motif enrichment analysis of different transcription factors (TFs) using nucleosome-depleted regions (NDRs) located in open chromatin regions and all identified nucleosome regions (NRs).

**Figure supplement 1—source data 1.** Source data for the enrichment scores of TFs.

**Figure supplement 2.** Comparison of the enrichment scores of pioneer factors for the maintenance of embryonic stem cell or reprogramming of somatic cells into induced pluripotent stem cells (Test set 3, highlighted as red) with other transcription factors (TFs).

**Figure supplement 3.** Enrichment analysis of transcription factors (TFs) by redefining differentially open regions as those closed in differentiated cell lines and open in H1 embryonic cell line.

**Figure supplement 3—source data 1.** Source data of TF enrichment scores calculated from the enrichment analysis by redefining differentially open regions as those closed in differentiated cell lines and open in H1 embryonic cell line.

**Figure supplement 4.** Transcription factor (TF) motif enrichment score is used to distinguish 11 known pioneer transcription factors (PTFs) with essential roles in cell differentiation (Test set 2) from other TFs.

**Figure supplement 4—source data 1.** Source data of TF enrichment scores used for ROC analysis.

**Figure supplement 5.** Transcription factor (TF) motif enrichment score is used to distinguish 11 known pioneer transcription factors (PTFs) with essential roles in cell differentiation (Test set 2) from other TFs.

**Figure supplement 5—source data 1.** Source data for TF enrichment scores calculated with different thresholds.

performance in classifications of PTF was not significantly affected by the threshold choice (*Figure 2—figure supplement 5*).

Finally, we performed an additional validation using recent data from the high-throughput protein microarray and electromobility shift assay experiments on human TFs which systematically assessed TF binding preferences to nucleosomal DNA versus naked DNA (*Fernandez Garcia et al., 2019*). The authors of this study classified TFs with respect to their strengths of nucleosome binding into three clusters: strong binders, which bind strongly to both naked and nucleosomal DNA (cluster 1), weak binders, which bind weakly to both naked and nucleosomal DNA (cluster 2), whereas cluster 3 consists of strong binders which bind strongly to naked DNA but weakly to nucleosomal DNA. We found that TFs from their cluster 3 had the lowest enrichment scores although this trend was not statistically significant because of a lack of the data in this cluster. As to TFs in clusters 1 and 2, half of them had binding sites enriched on NRs (including known PTFs FOXA1, GATA4, and CEBPA) which corresponded to strong nucleosome binders, and another half had binding sites enriched on NDRs (*Figure 2e*).

Using gene expression data from the Roadmap Epigenomics Program (*Bernstein et al., 2010*) and enrichment analysis on all open chromatin regions, we have identified 39 TFs in H1, K562, HepG2, and HeLa-S3 cell lines (no expression data was available for MCF-7) that ranked highly in the enrichment analysis and were also significantly expressed in the corresponding cell lines (*Supplementary file 1—table 2*). Among these 39 TFs, 15 TFs were well-characterized PTFs such as GATA, FOXA, and CEBP factors, ESRRB, NEUROD1, SPI1, and subunits of the AP-1 complex. To validate the remaining 20 PTF predictions, we performed literature searches and found that RFX5 had the ability to displace nucleosomes (*Zhu et al., 2018*). In addition, many of the predicted PTFs, such as ZKSCAN1, USF1, USF2, and SRF, were confirmed by a recent study (*Pop et al., 2023*).

Next, using the enrichment score calculated based on NRs located in differentially open chromatin regions and NDRs located in conserved open chromatin regions, we identified 40 TFs that could act as PTFs with essential roles in cell differentiation in K562, HepG2, and HeLa-S3 cell lines. These identified TFs had their DNA binding sites significantly enriched on nucleosomal DNA of differentially open chromatin regions and were significantly expressed in the corresponding cell lines (*Supplementary file 1—table 4*). Among these 40 TFs, 15 were well-characterized PTFs including GATA, FOXA, and CEBP factors and subunits of AP-1 complex. For the remaining 25 PTF predictions, 7 TFs were annotated in the literature as potential PTFs and/or potential nucleosome binders (*Supplementary file 1—table 8*). For instance, HNF4A was annotated as a potential PTF active in chromatin remodeling in the liver (*Qu et al., 2021*), LEF1 was identified as a regulatory high mobility group box protein that could bind to nucleosomes (*Steger and Workman, 1997*), and CUX1 could specifically interact with its recognition motif in a nucleosomal context (*Last et al., 1999*).

## Association between binding motif profiles and nucleosome occupancy for PTFs

The enrichment analysis described above is the first step to estimate the propensity of TF binding sites to be located on nucleosomal footprints. However, the enrichment analysis might have high specificity but a compromised sensititivity, as it can misclassify those PTFs that can bind to both naked DNA and nucleosomes at similar concentrations. Therefore, the next step would be to evaluate the actual locations of TF binding sites with respect to the nucleosomal dyad. To this end, we tested if there is a significant association between the binding motif profiles (see Materials and methods) and nucleosome occupancy values ±400 bp around the dyad (*Figure 3a*). Our results showed that 87% of all 225 TFs had negative PCC between binding motif profiles and nucleosome occupancy values (*Figure 3b*) which is consistent with the fact that nucleosomes generally restrict the access of TFs to their binding sites on DNA molecules.

We have also identified 37 TFs that had a statistically significant positive correlation coefficients between binding motif profile and nucleosome occupancy and could be classified as potential nucleosome binders and PTFs (*Supplementary file 1—table 9* and *Figure 3*). Among these predictions there were six known PTFs, such as POU5F1 (OCT4), GATA3, CEBPB, ATF2, NFYA, and NFYB (*Supplementary file 1—table 9*). As we mentioned previously, NFYB, NFYA, and POU5F1 had low enrichment scores, but were identified as nucleosome binders by this correlation analysis. It could be explained by their dual nature: these factors can bind to nucleosomes using certain binding arrangements, as

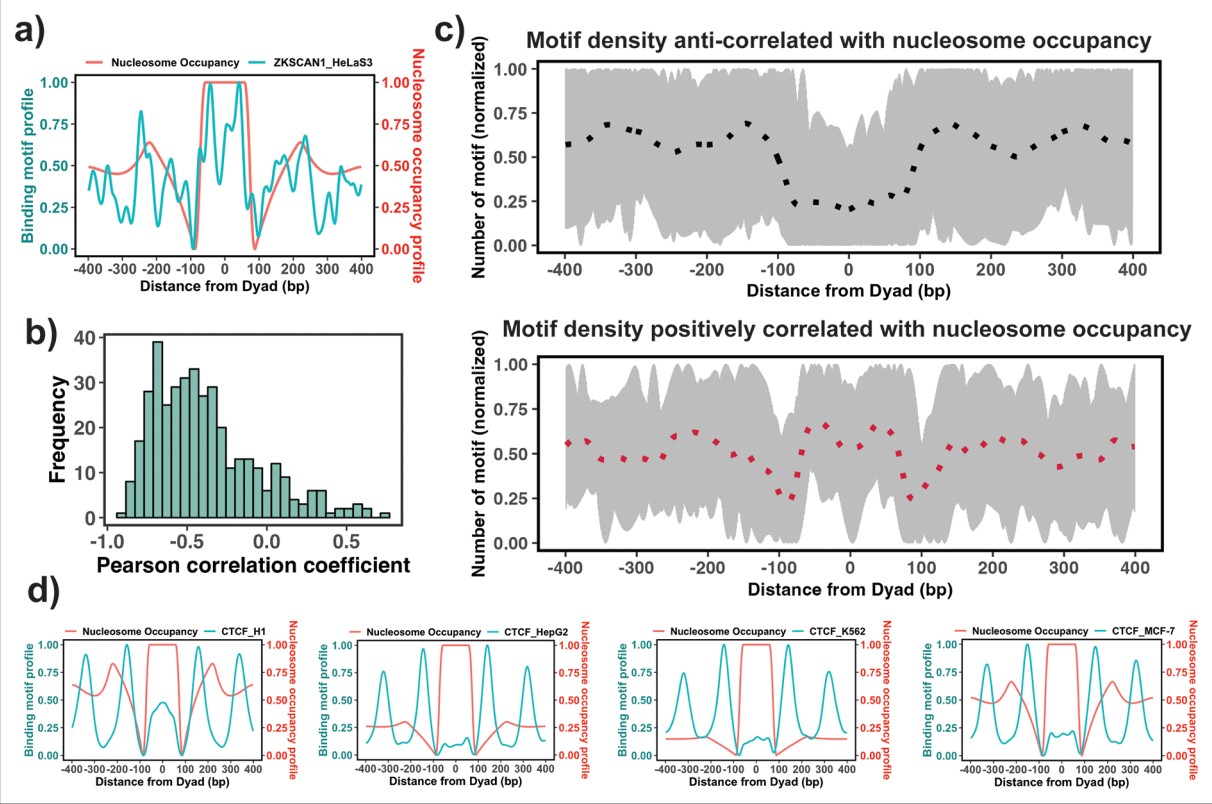

**Figure 3.** Association between the binding motif profiles and nucleosome occupancy. (**a**) Motif profiles for ZKSCAN1 in HeLa-S3 cell line is shown as an example. (**b**) Pearson correlation coefficients between motif profiles and nucleosome occupancy values for each transcription factor (TF) (n=225) (the median value of correlation coefficient = –0.46). (**c**) Binding motif profiles of TFs with positive (red, correlation coefficient ≥ 0.2and p-value <0.05) or negative correlation coefficients (black, correlation coefficient ≤ –0.4and p-value <0.05) between binding motif profile and nucleosome occupancy. Dashed lines correspond to the average of binding motif profiles. (**d**) Comparison of binding motif profiles of CTCF between H1 embryonic stem cell line and somatic cell lines (normalized within the range between 0 and 1).

The online version of this article includes the following source data for figure 3:

**Source data 1.** Source data for TF binding profiles, nucleosome occupancy levels and results of the correlation analysis.

evident from their positive correlation coefficients, but at the same time, they can interact with the naked DNA, as evident from their low enrichment scores. *Supplementary file 1—table 9* shows 16 predictions that coincided with the predicted PTFs from the enrichment analysis (ATF2, BACH1, CEBPB, CTCF, ESRRA, HMBOX1, NFATC3, NFYB, SREBF2, USF1, USF2, ZNF24, ZNF274, ZNF282, ZNF460 and ZKSCAN1). Among those with high PCC>0.5, there were ZKSCAN1, ESR1, NFATC3, ZBTB7B, MAX, and TBX2. Half of them were also predicted to be pioneers by a recent study (*Pop et al., 2023*). With the exception of a few cases, none of TFs was identified as being pioneer in all cell types because pioneer activity is often cell-type-specific. For example, CTCF was identified as having a significant correlation (although low) between binding motif profile and nucleosome occupancy for the human embryonic stem cell line (PCC = 0.1, p-value <0.05), but not for other somatic cell lines (*Figure 3d*). Indeed, previous studies have indicated that CTCF proteins could access the binding sites in nucleosomes and may function as PTFs in embryonic stem cells and to a lesser extent in differentiated cells (*Teif et al., 2012*; *Teif et al., 2014*; *Voong et al., 2016*).

## Deciphering the interaction modes between TFs and nucleosomes

A recent study characterized the interaction landscape between PTFs and nucleosomes using NCAP-SELEX (Nucleosome Consecutive Affinity-Purification with Systematic Evolution of Ligands by Exponential Enrichment) (*Zhu et al., 2018*). It revealed different binding modes of PTFs: DNA end binding, dyad binding, gyre binding, and periodic binding (*Zhu et al., 2018*). The NCAP-SELEX approach was based on the analysis of enrichment of specific sequences from the DNA libraries. These sequences

were reconstituted into nucleosomes and incubated with TFs (*Zhu et al., 2018*). Then, the dissociated nucleosomal DNA was separated from the intact nucleosomes and the analysis of the enrichment of sequences allowed to identify TF binding specificities and binding site locations on nucleosomal DNA (*Zhu et al., 2018*). To compare our TF motif profiles with NCAP-SELEX data, we filtered out low-quality motif profiles using the criteria described in the Materials and methods section and then calculated motif enrichment scores for TFs with different binding modes identified by NCAP-SELEX approach. Due to the limited number of TFs observed in both our dataset and NCAP-SELEX study (24 TFs), we mainly focused on the DNA end binding superhelical locations (SHLs) from ±5.5 to ±7 and the dyad binding modes with SHL from 0 to ±1.5. To estimate the preferential binding of TFs to the ends of nucleosomal DNA compared to the nucleosomal dyad, we calculated the end/dyad binding ratio ($R_{end/dyad}$) as the number of binding motifs at the DNA ends (SHLs from ±5.5 to ±7) divided by the number of binding motifs near dyad regions (SHLs from 0 to ±1.5).

In NCAP-SELEX experimental analyses, to quantify the preference of PTF binding to nucleosomes (*Zhu et al., 2018*), binding signals were compared by calculating the mutual information (MI) content between 3-mer distributions at two non-overlapping positions of the ligand, aimed at finding if SELEX ligand may contact these positions at the same time. Since nucleosomes can form on most DNA sequences, whereas TFs bind to only a few specific sequences, the NCAP-SELEX study calculated the enriched MI (EMI) score to separate the TF signals from nucleosome signals by limiting the MI measure to the top 10 most enriched 3-mer pairs. EMI penetration score corresponds to EMI drop by half compared to the EMI maximum and larger values pointed to the favorable binding to the dyad regions (*Zhu et al., 2018*). As our $R_{end/dyad}$ and experimental EMI penetration values should be anti-correlated, we indeed observed a statistically significant negative correlation between $R_{end/dyad}$ and EMI penetration values (PCC = –0.45, p-value = 0.025). This shows that our computational analysis

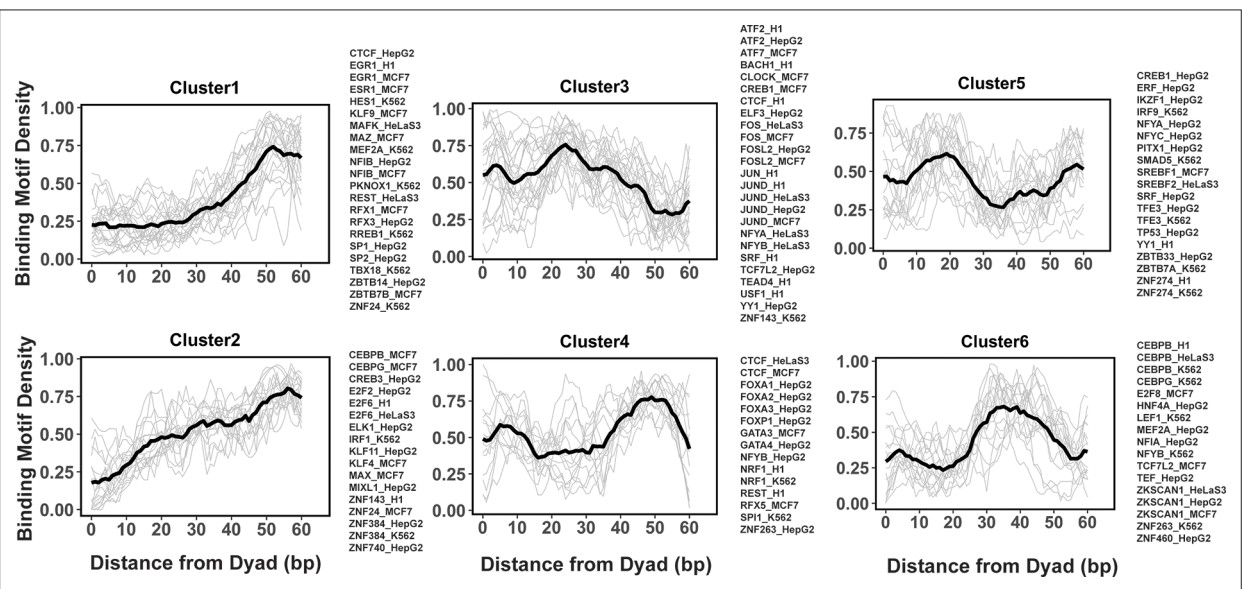

**Figure 4.** Clusters of transcription factor (TF) binding motif profiles on nucleosomal DNA. Binding motif profiles centered at nucleosomal dyad locations (±60bp from dyad) are clustered using *k*-medoids clustering with *k*=6. The entry/exit regions of nucleosomal DNA were excluded as a certain nucleotide bias exists around the ends of nucleosomal DNA reads produced by the MNase-seq experiments. Binding motif profiles between two symmetrical nucleosomal halves are combined for each TF. The black line represents the averaged profiles of all TFs in the same cluster. Cluster members with silhouette width ≤ 0.25 were considered as outliers and removed.

The online version of this article includes the following source data and figure supplement(s) for figure 4:

**Source data 1.** Source data for TF binding profiles in each cluster.

**Figure supplement 1.** Clusters analysis of TF binding motif profiles.

**Figure supplement 2.** A liner regression model describing the relationship between calculated transcription factor (TF) end/dyad binding ratio ($\tau_{end/dyad}$) from our study and the enriched mutual information (EMI) intensity and EMI penetration values from recent nucleosome NCAP-SELEX experiments (*Zhu et al., 2018*).

**Figure supplement 2—source data 1.** Source data of the EMI intensity and EMI penetration values from nucleosome NCAP–-SELEX experiments.

is able to capture the detailed features of TF binding motifs on nucleosomes. For the EMI intensity, although a negative linear dependence trend was evident, the correlation was not significant (*Figure 4—figure supplement 2*).

## Clustering analysis reveals several groups of PTF binding sites

Previous studies indicated that while nucleosome positioning acted as a barrier to TF binding, there was a number of different signatures of mutual TF/nucleosome positioning (*Kundaje et al., 2012*). We attempted to characterize the details of PTF binding motifs at near-single nucleotide resolution on nucleosomal DNA and performed *k*-medoids clustering of binding motif profiles for all 225 TFs (*Figure 4* and *Figure 4—figure supplement 1*). TFs in the first and second cluster showed that motif density increased with the distance from the dyad. These clusters include cases of canonical factors or those PTFs that preferentially interact with the ends of nucleosomal DNA from SHLs ±5 to SHL ±6. In clusters 4 and 6, TFs preferentially interact with the nucleosomal DNA around the dyad regions and SHL from ±3.5 to SHL ±5.5, whereas TFs from clusters 3 and 5 preferentially occupy SHL 1 to SHL ±3, respectively. Indeed, it has been recently shown that GATA3 factor targets nucleosomal DNA around the SHL ±5.5 position (*Tanaka et al., 2020*).

We found that many TFs from the same family and even from different cell types were assigned to the same cluster. For example, all known PTFs from FOXA families belong to cluster 4 and indeed it has been shown that FOXA1 can target its binding motifs to nucleosomal DNA near the dyad and linker regions (*Iwafuchi-Doi et al., 2016*). However, there are a few exceptions, such as CEBPB, CREB1, CTCF, MEF2A, NFYA, and NFYB, where the same TF from different cell lines has been assigned to different clusters.

## Discussion

The modulation of chromatin accessibility with high spatiotemporal precision is a subject of ongoing debate as it is a crucial factor in regulation of transcription, replication, and DNA repair processes. Chromatin accessibility is precisely modulated involving multiple aspects: epigenetic modifications, PTF infiltration, ATP-dependent remodeling, and spontaneous chromatin dynamics. There are about 2000 TFs in the human genome and a few hundred of them may have PTF properties. Yet, for the vast majority of human PTFs, the locations of their binding sites and mechanisms of binding and regulation remain unknown (*Fernandez Garcia et al., 2019*; *Lambert et al., 2018*). There are many different experimental assays that can provide information on TF binding sites but these assays suffer from multiple drawbacks and often require prior knowledge of the TFs being tested. Moreover, PTFs are very dynamic, may target partial binding sites, and work cooperatively with other factors. For example, two recent cryo-EM structures of the PTF SOX2-nucleosome complexes showed different mechanisms of binding (*Michael et al., 2020*; *Dodonova et al., 2020*). All this complicates the identification and characterization of human PTFs pointing to the pressing need to develop predictors and classifiers (*Zaret and Mango, 2016*; *MacCarthy et al., 2022*; *Zaret et al., 2016*; *Nagaich et al., 1999*).

The goal of this study has been to gain functional insights into the mechanisms of binding and infiltration of PTFs into chromatin at the level of nucleosomes. To achieve this goal, we used ChIP-seq, MNase-seq, and DNase-seq data from five different cell lines for 225 human TFs. A computational framework has been developed to systemically investigate the ability of TFs to bind to nucleosomes. As a result, we found that using the information on differentially open chromatin regions (open in one cell line, closed in another) leads to the highest classification accuracy in discriminating pioneer from canonical TFs. This finding supports the view that TF binding to nucleosomes leads to DNA and chromatin opening and might correlate with the reprogramming potential (*Fernandez Garcia et al., 2019*). Our study has also verified the known and predicted several dozens of new PTFs as nucleosome binders (*Supplementary file 1—tables 2, 4, and 9*). These predicted cases include TFs without previously known pioneer activity which could be subject to the future experimental validation. Finally, we have identified six distinctive clusters of TF binding profiles with the nucleosomal DNA. These clusters point to the diversity of binding motifs where TFs belonging to the same cluster may exhibit potential competitive binding.

We should mention that our classification method and the data used in this study have certain limitations. First, interactions of PTFs with nucleosomes may depend on binding of other factors.

Second, PTF binding relies on specific recognition of DNA binding sites and can be governed by nucleosome dynamics. Indeed, initial stages of binding can happen via the DNA site exposure through thermal fluctuations, leading to DNA unwrapping and the formation of DNA looping or twist defects (*Poirier et al., 2009*; *Li et al., 2022*; *Brandani et al., 2018*). PTFs can exploit DNA unwrapping and trap nucleosomes in a partially unwrapped states (*Donovan et al., 2019*). TF binding motifs close to the nucleosome entry-exit sites may have increased exposure but would be error-prone to capture using MNase-seq data as a nucleotide bias exists around the ends of nucleosomal DNA reads produced by the MNase-seq experiments. Moreover, as we showed recently, the degree of spontaneous DNA unwrapping from nucleosomes depends on the incorporation of histone variants, like H2A.Z (*Li et al., 2023*), which, in turn, might be needed for PTF recruitment to promote embryonic stem cell differentiation (*Li et al., 2012*). The third group of limitations pertains to the nucleosome positioning in a genome, as binding dissociation constants for PTFs depend on where nucleosomes are located. Indeed, nucleosome positioning and stability is not uniform throughout the genome and depends on local nucleosomal DNA sequence, histone variant deposition, and epigenetic chromatin modifications. Finally, PTFs may exhibit multivalent binding recognizing not only DNA binding sites but also some parts of the histone core or histone tail regions. Deducing such dependencies from the experimental assays utilized in this study is very challenging, if not impossible.

Nucleosomes represent hub points in epigenetic signaling pathways and identifying complex epigenetic relationships at the level of single nucleosomes may yield functional insights into the mechanisms of binding and infiltration of PTFs into chromatin during differentiation and reprogramming. TFs regulate a large number of signaling pathways and their dysregulation contributes to a plethora of human diseases, including diabetes, cardiovascular diseases, and many cancers. Conventional TFs have been used for a long time as biomarkers and drug targets, however, the targeted therapeutic potential of PTFs is lagging. To fill this gap, integrative approaches using large-scale low- or medium-resolution data with precise molecular modeling or protein-protein docking may provide the required detailed characterization of many predicted PTFs in the future.

# Materials and methods

## Key resources table

| Reagent type (species) or resource | Designation | Source or reference | Identifiers | Additional information |
|---|---|---|---|---|
| Software, algorithm | nf-core-mnaseseq pipeline | *The Bioinformatics & Biostatistics Group and Patel, 2020* | | https://github.com/nf-core/mnaseseq |
| Software, algorithm | Burrows-Wheeler Aligner (BWA) | *Li and Durbin, 2009a* | | https://bio-bwa.sourceforge.net |
| Software, algorithm | BEDTools suite | *Quinlan and Hall, 2021* | | https://bedtools.readthedocs.io/en/latest/content/bedtools-suite.html |
| Software, algorithm | FIMO | *Grant et al., 2011a* | | https://meme-suite.org/meme/doc/fimo.html |
| Software, algorithm | t-SNE | *Donaldson, 2022* | | https://cran.r-project.org/web/packages/tsne/index.html |
| Software, algorithm | Cluster | *Maechler et al., 2023* | | https://cran.r-project.org/web/packages/cluster/index.html |
| Software, algorithm | TTR | *Ulrich and Smith, 2023* | | https://cran.r-project.org/web/packages/TTR/index.html |

## Genome-wide mapping of nucleosome dyads and footprint regions

High-coverage micrococcal nuclease sequencing data (MNase-seq) of five human cell lines (H1, HepG2, MCF-7, K562, and HeLa) from paired-end sequencing was used for nucleosome mapping (*Supplementary file 1—table 10*). The raw MNase-seq data was downloaded from the NCBI Sequence Read

Archive (SRA) and converted into the fastq format using SRA Toolkit (*Leinonen et al., 2011*). Then, the downloaded fastq files were processed using the mnaseseq pipeline from nf-core (*The Bioinformatics & Biostatistics Group and Patel, 2020*; *Ewels et al., 2020*), a recently developed bioinformatics pipeline for MNase-seq data analysis. Adapter trimming of sequencing reads was performed with Trim Galore. Then, adapter-trimmed reads were mapped to the reference genome using Burrows-Wheeler Aligner (BWA) (*Li and Durbin, 2009b*). Human genome GRCh37 was used as a reference genome for reads mapping and the maximum number of mismatches in alignment was set to 4. The minimum and maximum insert sizes for filtering of mono-nucleosome paired-end reads were set to 120 and 180 bp. Duplicate reads were marked using Picard MarkDuplicates command (http://broadinstitute.github.io/picard/) and discarded from the analysis to avoid PCR duplication artifacts (less than 10% of reads were duplicated). Read libraries of replicates from the same experiment condition were merged into the analysis. Next, the BAM sequence alignment files were converted into BED format using bedtools v2.30.0 (*Quinlan and Hall, 2010*). The aligned reads with fragment sizes from 146 to 148 bp were selected for mapping of the representative dyad positions (center of the nucleosomal DNA).

To determine the representative dyad position of nucleosomes, we implemented and modified a previously developed nucleosome mapping protocol (*Valouev et al., 2011*) and dyad positions were determined as midpoints of mapped MNase-seq reads (*Gaffney et al., 2012*). Namely, we used a triweight kernel *K* as a weighting function (*Equation 1*) and the kernel-smoothed dyad counts *D* at each genomic coordinate *i* is calculated as (*Valouev et al., 2011*)

$$K(u) = \begin{cases} \left(1 - \left(\frac{u}{h}\right)^2\right)^3 & |u| \leq h \\ 0 & |u| > h \end{cases} \tag{1}$$

$$D(i) = \frac{1}{h} \sum_{j=1}^{N} K(i-j) \cdot d(j) \tag{2}$$

Here, *N* is the length of a chromosome, *d(j)* is the dyad count at genomic coordinate *j*, and *D(i)* is the smoothed dyad count at genomic coordinate *i*. Small values of bandwidth *h* lead to less smoothing but more accurate estimates of dyad positions. In our case, we chose a relatively small bandwidth value *h*=15 to improve the accuracy of the mapped dyad positions.

Next, we identified the genomic locations with the local maximum values of the smoothed dyad counts using bwtool (*Pohl and Beato, 2014*) and the minimum distance between the neighboring local maxima was set to 150 bp with 'find local-extrema -maxima -min-sep=150'. Then, for every 60 bp window centered at each local maxima, one representative dyad location was determined as the dyad location with the highest number of dyad counts in this interval. Other dyad positions within the same 60 bp interval were discarded. If two or more dyad positions had the same dyad counts within the same interval, the dyad position located closest to the local maximum of the smoothed counts was selected as the representative dyad. Last, NRs were determined as genomic regions centered at the representative dyad positions and flanked by 73 bp segments on each side.

## Genome-wide mapping of active enhancers, open chromatin, and NDRs

To map the open chromatin and active enhancer regions, we used DNase-seq and H3K27ac and H3K4me1 ChIP-seq data from the following five human cell lines (H1, HepG2, MCF7, K562, and HeLa-S3, *Supplementary file 1—tables 10 and 11*). The narrowPeak files were downloaded from the ENCODE and Roadmap project (*Moore et al., 2020*; *Kundaje et al., 2015*). The open chromatin regions were identified as genomic regions centered at narrow peaks and flanked by 1000bp segments on each side. The active enhancer regions were defined as the open chromatin regions overlapped with both H3K27ac and H3K4me1 ChIP-seq narrow peaks. The NDRs represent genomic regions free of nucleosomes and were identified as open chromatin regions not covered by any mono-nucleosome fragments (120–180 bp length) in MNase-seq data in all replicate experiments.

Using open chromatin regions from the DNase-seq data, we identified *differentially* and *conserved open chromatin* regions using the 'intersect' command from the BEDTools suite (*Quinlan and Hall, 2010*). Conserved open chromatin regions represent open chromatin regions that are more than 80% shared between embryonic H1 and at least one other differentiated cell line used in this study (*Piroeva et al., 2023*; *Figure 1—figure supplement 1*). Chromatin regions with differential

accessibility ('*differentially open chromatin regions*') are defined as those that have less than 20% overlap between open regions in H1 embryonic cell line and open regions in one of differentiated cell lines so that these regions are closed in one cell line type and open in another (*Piroeva et al., 2023*; *Figure 1—figure supplement 1*). While predicting PTFs, that are important during the differentiation of embryonic stem cells (TFs in Test set 2), we defined differentially open chromatin regions as those closed in H1 and open in differentiated cell lines. Vice versa, based on their functions, for seven factors from Test set 3, which act in reprogramming somatic cells into induced pluripotent stem cells, we defined the differentially open regions as those closed in differentiated cell lines and open in the H1 embryonic cell line. For motif enrichment analysis of TFs, we selected two sets of NRs and NDRs: (1) NDRs located in the open chromatin regions and all identified NRs using the MNase-seq data; (2) NDRs located in conserved open chromatin regions and NRs located in the *differentially* open regions so that their accessibility may be associated with TF binding.

## Analysis of dinucleotide patterns of nucleosomal DNA

Using identified representative nucleosome dyad positions, we examined dinucleotide patterns of nucleosomal DNA and mapped genomic coordinates of WW/SS (where W is A or T, and S is G or C) and YY/RR (R=A or G, and Y=C or T) dinucleotides on NRs. Then, we aligned NRs by superimposing their dyad positions and computed the frequency of observed dinucleotides at each location of nucleosomal DNA (as a function of distance in base pairs from the nucleosome dyad). As a result, we observed pronounced dinucleotide patterns at specific nucleosomal DNA positions for all cell lines which are indicative of the high quality of the data and nucleosome mapping procedure (*Figure 1—figure supplement 2*).

## Genome-wide mapping of TF binding sites

To map the genome-wide locations of binding sites of various TFs, we matched the ChIP-seq data from the ENCODE project (*Moore et al., 2020*) with the MNase-seq data for the same human cell lines (H1, HepG2, MCF-7, HeLa-S3, and K562) (*Supplementary file 1—table 12*). In case of HeLa cell line, we used ChiP-seq data in HeLa-S3, which is a clonal derivative of HeLa. In total, ChIP-seq data for 225 TFs could be matched with the corresponding MNase-seq data from the same cell type. All available narrow peak files of ChIP-seq of these TFs were downloaded, and files corresponding to the same TF were merged for further analysis. To map binding sites from ChIP-seq narrow peaks, referred to as 'ChIP-seq TF motif', we downloaded position frequency matrices (PFMs) for each TF from the JASPAR CORE database (*Sandelin et al., 2004*). Then, we applied FIMO program (*Grant et al., 2011b*) to scan the DNA sequences within ChIP-seq narrow peaks using each PFM and identified motifs with p-value less than $10^{-4}$ for further analysis.

Next, we calculated the TF binding motif profiles or briefly '*motif profiles*'. To do that, we first mapped ChIP-seq TF motifs for a given TF to the closest nucleosome dyad position and then aligned all dyad positions and corresponding ChIP-seq TF motifs. For each TF binding motif, we considered sequences in both strands of DNA. Then, we counted the number of base pairs from different TF binding motifs that mapped at each location of nucleosomal DNA and flanking DNA regions (±1000 bp) around the dyad.

## TF motif enrichment analysis

PTFs can engage nucleosomal DNA, while truly canonical TFs cannot bind to nucleosomes. To predict PTFs, we calculated the binding motif enrichment of different TFs on NRs compared to NDRs. Namely, we counted a number of base pairs of ChIP-seq TF motifs overlapped with the NRs and NDRs and constructed the following contingency table (*Supplementary file 1—table 13*). We then calculated the enrichment score for each TF (*Equation 3*) and applied a Fisher's exact test to evaluate the significance of TF motif enrichment on NRs compared to NDRs.

$$\textit{Enrichment score (Odds ratio)} = (a/c)/(b/d) \tag{3}$$

Here, *a* and *c* are the numbers of base pairs overlapped with ChIP-seq TF motifs on NR and NDR, respectively. Counts *b* and *d* correspond to the number of base pairs outside of ChIP-seq TF motifs on NR and NDR, respectively. We further collected the data on expression levels of various TFs from

the NIH roadmap epigenomics program and identified TFs which are significantly expressed in each cell line (RPKM value ≥ 10).

ROC and PR curve analyses were further performed to evaluate the power of enrichment scores in discriminating PTFs from other TFs. 32 known PTFs from the literature (which were also included in our list of 225 TFs) were used as positives (*Supplementary file 1—table 5*) and other TFs were considered as negatives for this test.

## Clustering of TF binding motif profiles

To identify the prevalent interaction modes of various TFs with nucleosomes, we have performed clustering of TF binding motif profiles. Prior to clustering, we filtered out potentially low-quality binding profiles using the following criteria. First, we removed under-represented TFs with the total genome-wide cumulative sum of ChiP-seq TF motifs on NRs of less than 500 bp. Second, due to the twofold symmetry of DNA in nucleosome structures, ChiP-seq TF motifs on DNA complementary plus and minus strands should be structurally superimposed if the nucleosome structure is rotated by 180 degrees. If we are analyzing relatively large number of binding motifs on both DNA strands, binding motif profiles should be symmetrical with respect to the nucleosome dyad. Therefore, we calculated the Pearson correlation coefficient of motif profiles between two symmetrical nucleosomal halves (positive and negative SHLs of nucleosomal DNA) for each TF and removed those with Pearson correlation coefficient values less than 0.4.

Next, we applied t-distributed stochastic neighbor embedding (t-SNE) and projected all profiles onto two dimensions using the Rtsne function from the R package (*van der Maaten, 2014*). Then, the projected data were subjected to *k*-medoids clustering using the pam function from the R package with the number of clusters equal to 6 (*Figure 4—figure supplement 1*). The silhouette width is an estimate of the goodness of clusters, its values close to 1 correspond to a cluster where most objects are much closer to other objects in the same cluster than to other clusters. For each cluster, members with silhouette width ≤ 0.25 were considered as outliers and 33 outliers were removed.

Since Micrococcal Nuclease has a sequence bias and cleaves DNA upstream of A or T more efficiently than of G or C nucleotides, a certain nucleotide preference exists around the ends of nucleosomal DNA reads produced by the MNase-seq experiments. It may potentially bias the TF binding profiles near the nucleosomal DNA ends. Therefore, in our analysis, we excluded regions near the nucleosomal DNA ends. To identify binding modes between TFs and nucleosomes, we clustered binding motif profiles of different TFs centered at the nucleosomal dyad locations (±60 bp from dyad) using *k*-medoids.

## Acknowledgements

ARP was supported by the Department of Pathology and Molecular Medicine, Queen's University, Canada. ARP is the recipient of a Senior Canada Research Chair in Computational Biology and Biophysics and a Senior Investigator Award from the Ontario Institute of Cancer Research, Canada. ARP acknowledges the support of the Natural Sciences and Engineering Research Council of Canada (NSERC) (No. RGPIN/02972-2021 ARP). ARP would like to thank David Clark for helpful discussions. YP was supported by the National Natural Science Foundation of China (No. 12205112) and Fundamental Research Funds for Central China Normal University. YP, WS, IO, and DL were supported by the Intramural Research Program of the National Library of Medicine, NIH. VBT acknowledges support by Cancer Research UK (grants EDDPMA-Nov21\100044 and SEBPCTA-2022/100001) and BBSRC IAA grant.

## Additional information

### Funding

| Funder | Grant reference number | Author |
| --- | --- | --- |
| Canada Research Chairs | Senior Canada Research Chair in Computational Biology and Biophysics | Anna R Panchenko |

| Funder | Grant reference number | Author |
| --- | --- | --- |
| Ontario Institute for Cancer Research | Senior Investigator Award | Anna R Panchenko |
| Natural Sciences and Engineering Research Council of Canada | RGPIN/02972-2021 | Anna R Panchenko |
| National Natural Science Foundation of China | No.12205112 | Yunhui Peng |
| Cancer Research UK Cambridge Institute, University of Cambridge | EDDPMA-Nov21\100044 and SEBPCTA-2022/100001 | Vladimir B Teif |
| National Institutes of Health | National Library of Medicine | Wei Song |

The funders had no role in study design, data collection and interpretation, or the decision to submit the work for publication.

### Author contributions

Yunhui Peng, Conceptualization, Data curation, Formal analysis, Validation, Investigation, Visualization, Methodology, Writing – original draft, Writing – review and editing; Wei Song, Vladimir B Teif, Methodology, Writing – review and editing; Ivan Ovcharenko, Investigation, Writing – review and editing; David Landsman, Supervision, Funding acquisition, Investigation, Writing – review and editing; Anna R Panchenko, Conceptualization, Supervision, Funding acquisition, Investigation, Visualization, Methodology, Writing – original draft, Project administration, Writing – review and editing

### Author ORCIDs

Yunhui Peng ⓘ https://orcid.org/0000-0001-9768-4127
Vladimir B Teif ⓘ http://orcid.org/0000-0002-5931-7534
David Landsman ⓘ https://orcid.org/0000-0002-9819-6675
Anna R Panchenko ⓘ https://orcid.org/0000-0003-3104-1131

Reviewer #1 (Public Review): https://doi.org/10.7554/eLife.88936.4.sa1
Reviewer #2 (Public Review): https://doi.org/10.7554/eLife.88936.4.sa2
Reviewer #3 (Public Review): https://doi.org/10.7554/eLife.88936.4.sa3
Author Response https://doi.org/10.7554/eLife.88936.4.sa4

# Additional files

### Supplementary files

- Supplementary file 1. Supplementary materials for this study.

- MDAR checklist

- Source code 1. Source codes for identifying nucleosome representative dyad positions.

- Source code 2. Source codes for processing MNase-seq raw data and mapping nucleosome dyad locations.

- Source code 3. Source codes for transcription factor (TF) enrichment score and binding motif profile analyses.

### Data availability

The datasets and computer code produced in this study are available from GitHub (copy archived at *Peng, 2023*) and additional data associated to this manuscript can be accessed on Zenodo.

The following dataset was generated:

| Author(s) | Year | Dataset title | Dataset URL | Database and Identifier |
| --- | --- | --- | --- | --- |
| Peng Y, Song W, Teif VB, Ovcharenko I, Landsman D, Panchenko AR | 2023 | Data for paper "Detection of new pioneer transcription factors as cell-type specific nucleosome binders" | https://doi.org/10.5281/zenodo.10418936 | Zenodo, 10.5281/zenodo.10418936 |

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
