## [Editor Report · eLife assessment]

This **valuable** study aims to identify pioneer transcription factors - which are defined as transcription factors that compete with nucleosomes for DNA binding. The authors provide methods for identifying pioneer transcription factors on a cell type basis, using nucleosome positioning and motif information across different cell lines. The evidence to support the claims is largely **solid**. This work will be of interest to computational and molecular biologists working on transcription factors.

---

## [Referee Report · Reviewer #1 (Public Review)]

Peng et al develop a computational method to predict/rank transcription factors (TFs) according to their likelihood of being pioneer transcription factors--factors that are capable of binding nucleosomes--using ChIP-seq for 225 human transcription factors, MNase-seq and DNase-seq data from five cell lines. The authors developed relatively straightforward, easy to interpret computational methods that leverage the potential for MNase-seq to enable relatively precise identification of the nucleosome dyad. Using an established smoothing approach and local peak identification methods to estimate positions together with identification of ChIP-seq peaks and motifs within those peaks which they referred to as "ChIP-seq motifs", they were able to quantify "motif profiles" and their density in nucleosome regions (NRs) and nucleosome depleted regions (NDRs) relative to their estimated nucleosome dyad positions. Using these profiles, they arrived at an odd-ratio based motif enrichment score along with a Fisher's exact test to assess the odds and significance that a given transcription factor's ChIP-seq motifs are enriched in NRs compared to NDRs, hence, its potential to be a pioneer transcription factor. They showed that known pioneer transcription factors had among the highest enrichment scores, and they could identify a number of relatively novel pioneer TFs with high enrichment scores and relatively high expression in their corresponding cell line. They used multiple validation approaches including (1) calculating the ROC-AUC and Matthews correlation coefficient (MCC) and generating ROC and precision-recall curves associated with their enrichment score based on 32 known pioneer TFs among their 225 TFs which they used as positives and the remaining TFs (among the 225) as negatives; (2) use of the literature to note that known pioneer TFs that acted as key regulators of embryonic stem cell differentiation had a highest enrichment scores; (3) comparison of their enrichment scores to three classes of TFs defined by protein microarray and electromobility shift assays (1. strong binder to free and nucleosomal DNA, 2. weak binder to free and nucleosomal DNA, 3. strong binding to free but not nucleosomal DNA); and (4) correlation between their calculated TF motif nucleosome end/dyad binding ratio and relevant data from an NCAP-SELEX experiment. They also characterize the spatial distribution of TF motif binding relative to the dyad by (1) correlating TF motif density and nucleosome occupancy and (2) clustering TF motif binding profiles relative to their distance from the dyad and identifying 6 clusters.

The strengths of this paper are the use of MNase-seq data to define relatively precise dyad positions and ChIP-seq data together with motif analysis to arrive at relatively accurate TF binding profiles relative to dyad positions in NRs as well as in NDRs. This allowed them to use a relatively simple odds ratio based enrichment score which performs well in identifying known pioneer TFs. Moreover, their validation approaches either produced highly significant or reasonable, trending results.

The weaknesses of the paper are relatively minor, and the authors do a good job of describing the limitations of the data and approach.

---

## [Referee Report · Reviewer #2 (Public Review)]

In this study, the authors utilize a compendium of public genomic data to identify transcription factors (TF) that can identify their DNA binding motifs in the presence of nuclosome-wrapped chromatin and convert the chromatin to open chromatin. This class of TFs are termed Pioneer TFs (PTFs). A major strength of the study is the concept, whose premise is that motifs bound by PTFs (assessed by ChIP-seq for the respective TFs) should be present in both "closed" nucleosome wrapped DNA regions (measured by MNase-seq) as well as open regions (measured by DNAseI-seq) because the PTFs are able to open the chromatin. Use of multiple ENCODE cell lines, including the H1 stem cell line, enabled the authors to assess if binding at motifs changes from closed to open. Typical, non-PTF TFs are expected to only bind motifs in open chromatin regions (measured by DNaseI-seq) and not in regions closed in any cell type. This study contributes to the field a validation of PTFs that are already known to have pioneering activity and presents an interesting approach to quantify PTF activity.

For this reviewer, there were a few notable limitations. One was the uncertainty regarding whether expression of the respective TFs across cell types was taken into account. This would help inform if a TF would be able to open chromatin. Another limitation was the cell types used. While understandable that these cell types were used, because of their deep epigenetic phenotyping and public availability, they are mostly transformed and do not bear close similarity to lineages in a healthy organism. Next, the methods used to identify PTFs were not made available in an easy-to-use tool for other researchers who may seek to identify PTFs in their cell type(s) of interest. Lastly, some terms used were not defined explicitly (e.g., meaning of dyads) and the language in the manuscript was often difficult to follow and contained improper English grammar.

---

## [Referee Report · Reviewer #3 (Public Review)]

Peng et al. designed a computational framework for identifying pioneer factors using epigenomic data from five cell types. The identification of pioneer factors is important for our understanding of the epigenetic and transcriptional regulation of cells. A computational approach toward this goal can significantly reduce the burden of labor-intensive experimental validation.

The authors have addressed my previous comments.

The main issue identified in this re-review is based on the authors' additional experiments to investigate the reproducibility of the pioneer factors identified in the previous analysis that anchored on H1 ESCs.

The additional analysis that uses the other four cell types (HepG2, HeLa-S3, MCF-7, and K562) as anchors reveals the low reproducibility/concordance and high dependence on the selection of anchor cell type in the computational framework. In particular, now several stem cell related TFs (e.g. ESRRB, POU5F1) are ranked markedly higher when H1 ESC is not used as the anchor cell type as shown in Supplementary Figure 5.

Of note, the authors have now removed the shape labels that denote Yamanaka factors in Figure 2c (revised manuscript) that was presented in the main Figure 2a in the initial submission. The NFYs and ESRRB labels in Supplementary 4a are also removed and the boxplot comparing NFYs and ESRRB with other TF are also removed in this figure. Removing these results effectively hides the issues of the computational framework we identified in this revision. Please justify why this was done.

In summary, these new results reveal significant limitations of the proposed computational framework for identifying pioneer factors. The current identifications appear to be highly dependent on the choice of cell types.

---

## [Author Response]

The following is the authors’ response to the previous reviews.

**Public Reviews:**

**Reviewer #1 (Public Review):**
Peng et al develop a computational method to predict/rank transcription factors (TFs) according to their likelihood of being pioneer transcription factors--factors that are capable of binding nucleosomes--using ChIP-seq for 225 human transcription factors, MNase-seq and DNase-seq data from five cell lines. The authors developed relatively straightforward, easy to interpret computational methods that leverage the potential for MNase-seq to enable relatively precise identification of the nucleosome dyad. Using an established smoothing approach and local peak identification methods to estimate positions together with identification of ChIP-seq peaks and motifs within those peaks which they referred to as "ChIP-seq motifs", they were able to quantify "motif profiles" and their density in nucleosome regions (NRs) and nucleosome depleted regions (NDRs) relative to their estimated nucleosome dyad positions. Using these profiles, they arrived at an odd-ratio based motif enrichment score along with a Fisher's exact test to assess the odds and significance that a given transcription factor's ChIP-seq motifs are enriched in NRs compared to NDRs, hence, its potential to be a pioneer transcription factor. They showed that known pioneer transcription factors had among the highest enrichment scores, and they could identify a number of relatively novel pioneer TFs with high enrichment scores and relatively high expression in their corresponding cell line. They used multiple validation approaches including (1) calculating the ROC-AUC and Matthews correlation coefficient (MCC) and generating ROC and precision-recall curves associated with their enrichment score based on 32 known pioneer TFs among their 225 TFs which they used as positives and the remaining TFs (among the 225) as negatives; (2) use of the literature to note that known pioneer TFs that acted as key regulators of embryonic stem cell differentiation had a highest enrichment scores; (3) comparison of their enrichment scores to three classes of TFs defined by protein microarray and electromobility shift assays (1. strong binder to free and nucleosomal DNA, 2. weak binder to free and nucleosomal DNA, 3. strong binding to free but not nucleosomal DNA); and (4) correlation between their calculated TF motif nucleosome end/dyad binding ratio and relevant data from an NCAP-SELEX experiment. They also characterize the spatial distribution of TF motif binding relative to the dyad by (1) correlating TF motif density and nucleosome occupancy and (2) clustering TF motif binding profiles relative to their distance from the dyad and identifying 6 clusters.The strengths of this paper are the use of MNase-seq data to define relatively precise dyad positions and ChIP-seq data together with motif analysis to arrive at relatively accurate TF binding profiles relative to dyad positions in NRs as well as in NDRs. This allowed them to use a relatively simple odds ratio based enrichment score which performs well in identifying known pioneer TFs. Moreover, their validation approaches either produced highly significant or reasonable, trending results.The weaknesses of the paper are relatively minor, and the authors do a good job describing the limitations of the data and approach.
**Reviewer #2 (Public Review):**
In this study, the authors utilize a compendium of public genomic data to identify transcription factors (TF) that can identify their DNA binding motifs in the presence of nuclosome-wrapped chromatin and convert the chromatin to open chromatin. This class of TFs are termed Pioneer TFs (PTFs). A major strength of the study is the concept, whose premise is that motifs bound by PTFs (assessed by ChIP-seq for the respective TFs) should be present in both "closed" nucleosome wrapped DNA regions (measured by MNase-seq) as well as open regions (measured by DNAseI-seq) because the PTFs are able to open the chromatin. Use of multiple ENCODE cell lines, including the H1 stem cell line, enabled the authors to assess if binding at motifs changes from closed to open. Typical, non-PTF TFs are expected to only bind motifs in open chromatin regions (measured by DNaseI-seq) and not in regions closed in any cell type. This study contributes to the field a validation of PTFs that are already known to have pioneering activity and presents an interesting approach to quantify PTF activity.For this reviewer, there were a few notable limitations. One was the uncertainty regarding whether expression of the respective TFs across cell types was taken into account. This would help inform if a TF would be able to open chromatin. Another limitation was the cell types used. While understandable that these cell types were used, because of their deep epigenetic phenotyping and public availability, they are mostly transformed and do not bear close similarity to lineages in a healthy organism. Next, the methods used to identify PTFs were not made available in an easy-to-use tool for other researchers who may seek to identify PTFs in their cell type(s) of interest. Lastly, some terms used were not define explicitly (e.g., meaning of dyads) and the language in the manuscript was often difficult to follow and contained improper English grammar.
**Reviewer #3 (Public Review):**
Peng et al. designed a computational framework for identifying pioneer factors using epigenomic data from five cell types. The identification of pioneer factors is important for our understanding of the epigenetic and transcriptional regulation of cells. A computational approach toward this goal can significantly reduce the burden of labor-intensive experimental validation.The authors have addressed my previous comments.The main issue identified in this re-review is based on the authors' additional experiments to investigate the reproducibility of the pioneer factors identified in the previously analysis that anchored on H1 ESCs.The additional analysis that uses the other four cell types (HepG2, HeLa-S3, MCF-7, and K562) as anchors reveals the low reproducibility/concordance and high dependence on the selection of anchor cell type in the computational framework. In particular, now several stem cell related TFs (e.g. ESRRB, POU5F1) are ranked markedly higher when H1 ESC is not used as the anchor cell type as shown in Supplementary Figure 5.Of note, the authors have now removed the shape labels that denote Yamanaka factors in Figure 2c (revised manuscript) that was presented in the main Figure 2a in the initial submission. The NFYs and ESRRB labels in Supplementary 4a are also removed and the boxplot comparing NFYs and ESRRB with other TF are also removed in this figure. Removing these results effectively hides the issues of the computational framework we identified in this revision. Please justify why this was done.In summary, these new results reveal significant limitations of the proposed computational framework for identifying pioneer factors. The current identifications appear to be highly dependent on the choice of cell types.

Response: We thank all reviewers for their thoughtful and constructive comments and suggestions, which helped us to strengthen our paper. Following the suggestions, we have further addressed the reviewer’s comments and the detailed responses are itemized below.

**Reviewer #1 (Recommendations For The Authors):**
The following few minor mistakes/discrepancies/omissions should be addressed:1. In Figure 3, the Nucleosome Occupancy curves and legend are orange and the Binding Motif Profiles are blue; however, the y-axis label for Nucleosome occupancy profile is blue, and the y-axis label of Binding motif profile is orange. The colors seem to be switched, or I'm missing something.

Response: We thank the reviewer for pointing it out. We have changed the colors to make it consistent.

2. The text at the bottom of p. 11 of the main manuscript describing Supplementary Fig. 5 states: "If we repeat our anaysis by redefining differentially open regions as those closed in differentiated cell lines and open in H1 embryonic cell line, then ESSRB and Yamanaka pioneer transcription factor POU5F1 (OCT4) showed significantly higher enrichment scores (Supplementary Figure 5)." However, Supplementary Fig. 5 legend states: "Enrichment analysis of different TFs using the differentially open from one cell line (shown in the title) and conserved open regions from other four cell lines.". These two descriptions of the differential chromatin criteria used in the analysis don't appear to match. The description in the text is the one that makes much more sense to me. The legend should be written a little more clearly and reflect the statement in the main text. One can see from the cut and paste the "analysis" is also misspelled.

Response: We have rewritten the legend of Supplementary Figure 5 to make it clear and consistent. The misspelling has also been corrected.

3. It might be helpful to add that a random classifier would yield a constant precision recall (PR) curve (as a function of Recall) with the Precision = P/(P+N) or the fraction of positives for all plotted PR curves which in the case of Fig. 2a is 32/225 = 0.142, for example.

Response: We thank the reviewer for the suggestions. We have added the fraction of positives for Figure 2.

4. On p. 17 line 513, the authors refer to "Supplementary 7, 9 and 13". I'm assuming it's "Supplementary Tables 7, 9 and 13".

Response: It has been corrected.

5. On p. 18 line 539, "essays" should be "assays".

Response: It has been corrected.

**Reviewer #2 (Recommendations For The Authors):**
We are satisfied with the revisions in this version of the manuscript.
**Reviewer #3 (Public Review):**
The main issue identified in this re-review is based on the authors' additional experiments to investigate the reproducibility of the pioneer factors identified in the previously analysis that anchored on H1 ESCs.The additional analysis that uses the other four cell types (HepG2, HeLa-S3, MCF-7, and K562) as anchors reveals the low reproducibility/concordance and high dependence on the selection of anchor cell type in the computational framework. In particular, now several stem cell related TFs (e.g. ESRRB, POU5F1) are ranked markedly higher when H1 ESC is not used as the anchor cell type as shown in Supplementary Figure 5.Of note, the authors have now removed the shape labels that denote Yamanaka factors in Figure 2c (revised manuscript) that was presented in the main Figure 2a in the initial submission. The NFYs and ESRRB labels in Supplementary 4a are also removed and the boxplot comparing NFYs and ESRRB with other TF are also removed in this figure. Removing these results effectively hides the issues of the computational framework we identified in this revision. Please justify why this was done.In summary, these new results reveal significant limitations of the proposed computational framework for identifying pioneer factors. The current identifications appear to be highly dependent on the choice of cell types.

Response: We would like to clarify that our enrichment score used for TF classification, defined by Equation 3, is expected to be cell-type specific. The value of the enrichment score is modulated by a number of factors beyond the property of a TF to act as a PTF, such as the abundance of a given TF in a given cell line, cell type-specific nucleosome binding maps and interactions with other TFs. Thus, it is expected that the enrichment scores calculated for the same TF in different cell lines should be quantitatively different. Following the initial suggestion of Reviewer 3, we have diversified our analysis by using different cell lines as anchors. This analysis showed that most PTFs that we identified could be confirmed based on different cell lines, when comparing the relative enrichment scores within each cell line. On the other hand, it is not expected that the values of enrichment scores of a given TF should be similar across different cell lines.

Regarding a specific comment about ESRRB and POU5F1, these TFs are known pioneer factors with roles in reprogramming of somatic cells into induced pluripotent stem cells and suppressing cell differentiation. They have the ability to open closed chromatin regions in the differentiated cell lines. Therefore, if we redefine the differentially open regions as those closed in differentiated cell lines and open in H1 embryonic cell line, these pioneer factors are expected to have high enrichment scores. Indeed, our new results validated the roles of these PTFs in cell reprogramming. As mentioned above, their enrichment scores in different cell lines are not expected to be the same.

We also would like to clarify that no results were removed during the update of the figures, and all modifications of the manuscript following the suggestions of the reviewers were only made to improve the figures and make them clearer and the message more straightforward.